# A Point Cloud Simplification Algorithm Based on Weighted Feature Indexes for 3D Scanning Sensors

**DOI:** 10.3390/s22197491

**Published:** 2022-10-02

**Authors:** Zhiyuan Shi, Weiming Xu, Hao Meng

**Affiliations:** Department of Military Oceanography and Hydrography, Dalian Naval Academy, Dalian 116018, China

**Keywords:** 3D scanning sensors, point cloud simplification, feature index, bounding box, kd-tree, analytic hierarchy process, criteria importance through intercriteria correlation

## Abstract

Conventional point cloud simplification algorithms have problems including nonuniform simplification, a deficient reflection of point cloud characteristics, unreasonable weight distribution, and high computational complexity. A simplification algorithm, namely, the multi-index weighting simplification algorithm (MIWSA), is proposed in this paper. First, the point cloud is organized with a bounding box and kd-trees to find the neighborhood of each point, and the points are divided into small segments. Second, the feature index of each point is calculated to indicate the characteristics of the points. Third, the analytic hierarchy process (AHP) and criteria importance through intercriteria correlation (CRITIC) are applied to weight these indexes to determine whether each point is a feature point. Fourth, non-feature points are judged as saved or abandoned according to their spatial relationship with the feature points. To verify the effect of the MIWSA, 3D model scanning datasets are calculated and analyzed, as well as field area scanning datasets. The accuracy for the 3D model scanning datasets is assessed by the surface area and patch numbers of the encapsulated surfaces, and that for field area scanning datasets is evaluated by the DEM error statistics. Compared with existing algorithms, the overall accuracy of the MIWSA is 5% to 15% better. Additionally, the running time is shorter than most. The experimental results illustrate that the MIWSA can simplify point clouds more precisely and uniformly.

## 1. Introduction

The development of 3D scanning sensor technology, such as light detection and ranging (LiDAR), has gifted us with point cloud data with accurate coordinates together with other attributes [1], which enables it to play an increasingly important role in many fields such as road extraction [2], agriculture supervision [3], manufacturing monitoring [4], and terrain sensing [5]. However, the high-volume data that contain billions of points may cause great burdens in terms of data processing with regard to computing cost and storage space [6]. For this reason, it is necessary to simplify point clouds in order to alleviate the pressure [7]. This paper is intended to propose a solution to the simplification of point clouds obtained by scanning 3D models and field areas.

Numerous algorithms have been proposed for point cloud simplification. Depending on whether the point cloud is reconstructed, there are two types: mesh-based algorithms and point-based algorithms [8]. Usually, mesh-based algorithms require too many computing resources. Consequently, more attention has been paid to point-based simplification algorithms. In terms of the data attributes used for simplification, there are three categories: geometric features, spatial division, and extra features [9]. The extra feature algorithms do not yield enough information, and thus the first two kinds have been more broadly researched.

Simplification algorithms based on geometric features require the preservation of the global shape, the sharp areas, and some transition areas [10]. This can be achieved through evaluating the importance of each point via commonly defined feature indexes: curvature, density, distance, slope, and normal [6]. Moreover, some new feature indexes have been proposed in recent years. Direct Hausdorff distance is a measure that indicates how far two subsets of a metric space are from each other. Li et al. [11] used this measure to simplify point clouds. Markovic et al. [12] proposed a sensitive feature based on insensitive support vector regression.

Spatial division simplification algorithms commonly divide point cloud space into several sub-sets based on grid, octree, kd-tree, or other principles, or gather similar points into sub-clusters by k-means, the density-based spatial clustering of applications with noise (DBSCAN), etc. Some new point cloud division methods have also been devised. For example, Shoaib et al. [13] applied a fractal sense to a kind of 2D bubble point cloud division method. This kind of spatial division is able to relieve computational and memory stress and support a multi-scale operation.

It is reasonable to combine the above two categories: selecting feature points according to one or more geometric features and then resampling the points via spatial division. Additionally, some researchers have utilized more than one feature. Wang et al. [14] defined a comprehensive feature index including the average distance, normal, and curvature to distinguish feature points. After that, the non-feature points are simplified by uniform spherical sampling. Zhu et al. [15] set the importance level for point clouds by principal component analysis (PCA) and resampled them using different grid sizes. Zhang et al. [9] defined four kinds of entropy including scale-keeping, profile-keeping, and curve-keeping simplification to quantify the geometric features and then simplified the data under the consideration of neighborhoods. Ji et al. [16] calculated four feature indexes to extract feature points by describing the importance of points based on a space bounding box and then used octree to simplify the non-feature points. However, these methods either do not apply enough indexes to reveal the inner characteristics or do not consider the weight of multiple indexes. Furthermore, they do not take the relative position relationship between non-feature points and feature points into account. Some algorithms that extract the surface attributes of point clouds demand too much computation.

Thus, it can be concluded that some of the drawbacks of the existing point-based algorithms are: nonuniform simplification, the deficient reflection of point cloud characteristics by feature indexes, undiscussed weight distribution between feature indexes, and high computation cost. To deal with these problems, we propose a new algorithm named the multi-index weighting simplification algorithm (MIWSA). The MIWSA gives the following solutions to these problems:(1)Regarding the nonuniform simplification of point clouds, the MIWSA uses a bounding box and kd-trees to organize the points, which can precisely identify the relationship of the local points.(2)In order to achieve the deficient reflection of point cloud characteristics, the MIWSA applies five existing or newly created feature indexes to extract the characteristics of point clouds. More indexes enable the MIWSA to deal with different types of point clouds and to discover more inner information.(3)To solve the weight distribution problem, we use the improved analytic hierarchy process (AHP) and the criteria importance through intercriteria correlation (CRITIC) to reveal the relative importance of each index. These two weighting methods work based on data traits and human experience, respectively, which guarantees a more credible weight distribution scheme.(4)In order to reduce the computational complexity, the MIWSA makes some adjustments to the formulas. Moreover, the bounding box and kd-trees can also help improve its efficiency.

We remarkably promote point simplification effects in terms of both precision and efficiency with these improvements.

The rest of the paper is structured as follows. Section 2 looks at the research related to the MIWSA, including the point cloud organization method, the analysis of several point cloud features, and the AHP and CRITIC methods. After that, the procedures and key steps of MIWSA are elaborated in Section 3. Section 4 describes an experiment using several types of datasets and compares the results of different algorithms. Conclusions are drawn in Section 5.

## 2. Related Research

In the following section, we will illustrate the related studies that form the foundation of the MIWSA.

### 2.1. Cloud Organization

The MIWSA applies a bounding box and kd-trees to build the relationship effectively.

#### 2.1.1. Bounding Box

The bounding box method involves dividing the space into multiple small cubic grids and determining the grid where each point is located according to the three-dimensional coordinates so that it is easier to establish the neighborhood attribute between the points [17]. The construction of a bounding box involves the total point number N, the pre-set side length l of each voxel, and the point cloud volume V [18]. First, the maximum span of each dimension is calculated using Equation (1):(1)$$R_{i}=\max\left(G_{i}\right)-\min\left(G_{i}\right)$$
where $R_{i}$ means the point cloud span in the *i*th dimension, and $G_{i}$ means the point location value in each dimension. Then, the space volume of the point cloud can be expressed as:(2)$$V=\prod_{i=1}^{3}R_{i}$$

Since the side length L of the cubic grids is directly proportional to l and inversely proportional to N, the value of L is:(3)$$L={\left(\rho lV/N\right)}^{1/3}$$
where ρ is the adjustment parameter. The number of all the cubes is:(4)$$F=\prod_{i=1}^{3}\left\lceil R_{i}/L\right\rceil$$

#### 2.1.2. Kd-Tree

A kd-tree, which is mainly used for distance searches and nearest neighbor searches, is a spatial partition data structure based on a binary spatial partition tree. It can be used to search multidimensional spatial data. Xiao et al. [19] applied a kd-tree to represent the spatial topology relationship among sample points.

The space is split into two segments in each layer by a kd-tree. The top node splits the space into one dimension, and the next node splits the space into another dimension. This means that about half the points are stored in the left subtree and the other half are stored in the right subtree. When the number of points in a node is less than a certain value, it stops splitting and becomes a leaf node. The longest axis or the axis corresponding to the maximum variance is chosen as the splitting axis before each operation. The equation to calculate the variances is:(5)$$S_{G_{i}}^{2}=\frac{1}{N}{\sum_{k=1}^{N}\left(G_{i}-\frac{1}{N}\sum_{k=1}^{N}G_{i}\right)}^{2}$$
where S is the variance of each dimension and N is the total number of points in the point cloud. Then, the intermediate coordinate value of the splitting axis is commonly used as the splitting point.

A binary search method is used to find the distance range or the nearest data point, which enables a kd-tree to search for the nearest neighbor quickly and efficiently [20]. The complexity of creating kd-trees for all points is, respectively:(6)$$T_{c}=O\left(K\times N\times \log N\right)$$
and that of neighborhood depth-first search is:(7)$$T_{d}=O\left(N\times \log N\right)$$
where K is the number of dimensions. In a 3D point cloud simplification problem, K the value tends to be 3. According to Li et al. [21], the worst backtracking computational time complexity of a kd-tree is:(8)$$T_{w}=O\left(d\times N^{\left(2-1/K\right)}\right)$$

When K is limited and N exceeds a certain range, the change in logN will be very slight, and the time complexity will not increase significantly with an increase in N. The spatial division of three-dimensional point clouds meets this condition. Therefore, the application of a kd-tree to this kind of data can help to maintain high efficiency.

### 2.2. Point Cloud Features

In order to fully extract and quantize the features of point clouds, this paper uses or improves several point cloud feature indexes. According to a multivariable analysis [22], these indexes have the properties of relative independence, easy local calculation, and comprehensiveness. Below we give an introduction to former research on these existing indexes.

#### 2.2.1. Curvature Feature

This feature involves the extraction of sharp features. The most intuitive geometric attribute of scanned objects, which include corners, edges, and other structures, is the greater curvature. Therefore, the curvature-based index is used to analyze each point, and the points with greater curvature are retained [23]. Based on the basic concept of simplification by curvature, Wei et al. [24] further developed curvature co-occurrence histograms, and Wang et al. [25] proposed an adaptive curvature.

Generally, the image gradient method, PCA, triangular mesh method, or least squares surface fitting method is used to calculate the curvature [16,26]. Information on curvature is useful for describing the characteristics of points and can be used flexibly, but its computational complexity is generally high [27]. At the same time, the curvature method is very sensitive to noise points and does not have very good robustness, so it easily mischaracterizes noise points as feature points. Hence, the method of gaining curvature information is improved in this article.

#### 2.2.2. Density Feature

This feature involves the point cloud density. The density of scanning point clouds is irregular. Generally, the areas with denser point clouds have stronger characteristics, and the areas with low density have weaker characteristics [28]. The feature of point cloud density can accurately reflect the target spatial distribution and can be widely used in LiDAR data processing and applications. Point cloud density information cannot be used as the basis for point cloud simplification, but it can provide a reasonable reference for the importance evaluation of points. Consequently, considering the density feature index can help us to better retain detailed information on point clouds [14,28].

#### 2.2.3. Edge Feature

This feature involves preserving the object edge information. When scanning and collecting the model to obtain three-dimensional point clouds, the fact that the model scanning cannot be completed at one time is a problem. A model needs to be scanned many times, and then the many pieces of the point cloud are registered or spliced to output the complete three-dimensional point cloud of the model. The scattered point clouds obtained from a single scan are non-closed and have a large amount of edge information. These edge points contain important feature information for subsequent point cloud splicing, registration, simplification, and repair. The protection effect of edge points directly affects the results of the above operations. Therefore, the protection of boundary data points is very important. Qian et al. [29] used neighborhood points to construct a least squares surface, calculated the included angle between the neighborhood points and the projection line of the surface, and selected the boundary points according to the angle. Chen et al. [30] used a quick sequencing method to initially extract the edge points.

#### 2.2.4. Terrain Feature

This feature involves the effective simplification of terrain point clouds. The complexity of the terrain needs to be assessed. There are currently terrain indexes such as slope, fluctuation, geometry, and some other defined features [31,32]. In order to balance the speed and precision of the calculation, a new feature index considering the relevant height difference of the total points is proposed based on the former methods. This enhances the adaptivity of dealing with terrain point clouds.

### 2.3. The AHP and CRITIC Weighting Method

#### 2.3.1. The AHP Method

The analytic hierarchy process, firstly developed by T. L. Saaty in 1971–1975, is a general theory of measurement. It stipulates that the experience of people when making decisions is at least as valuable as data [33]. Based on this, it is used to give ratio scales to paired comparisons, which are derived from actual measurements [34]. It is concerned with measurements of both physical and psychological objectives. It has been widely applied in selection, evaluation, allocation, decision-making, etc. [35]. In this article, it is used to assess the relative importance of point cloud features and to give a proper weight determination scheme.

The main idea behind the AHP method is the deep analysis of complicated problems and their division into restrictive elements. During this process, a hierarchic or network structure is established. In general, such a structure might be one that descends from an overall objective, down to criteria, then down to the subdivisions of these criteria, and finally to the elements from which the choices are to be made. After that, paired comparisons are made to quantize the relative weights of the elements according to personal subjective judgments of their contribution based on a given criterion. After all the comparisons, the weights can be applied to any two elements in a process known as the principle of hierarchy composition.

The most basic step in AHP is to construct the judgment matrix. The matrix uses a certain scale to quantify people’s subjective judgment. It finally determines the results of evaluations, making it very important in the application of AHP. When constructing the judgment matrix, the scaling method proposed by T. L. Saaty is generally used to describe the relative importance relationship between the two elements, which applied labels such as equal importance, slight importance, obvious importance, strong importance, or extreme importance.

However, it is necessary to verify the consistency of the judgment matrix before using it. In the actual decision-making process, because the judgment matrix is easily affected by subjective preferences, it often cannot pass the consistency test, resulting in the failure of decision-making, which brings great difficulties to the practical application of AHP. In addition, if the number of elements to be compared is n, then n(n−1)/2 paired comparisons are made in the judgment matrix, resulting in a very large number of calculations. Therefore, the scale-extended method can be used to determine the judgment matrix [36]. In this way, no matter what scale is adopted, the constructed judgment matrix is completely consistent. Accordingly, there is no need to conduct a consistency test, and the calculation amount is significantly reduced.

The scale-extended method determines the judgment matrix by firstly sorting all the n indexes in the non-reduction order of importance as $x_{1}$, $x_{2}$, …, and $x_{n}$. After that, we use the scale values as the relative importance relationship between indexes, as shown in Table 1. After the relative importance between adjacent indexes is compared according to Table 1, the scale value between $x_{i}$ and $x_{i+1}$ is determined and recorded as $t_{i}$. Finally, the scale values between all adjacent indexes $t_{1}$, $t_{2}$, …, and $t_{n-1}$ are obtained. Other elements in the judgment matrix are obtained according to the transitivity of importance, then the final judgment matrix J can be written as:(9)$$J=\left[\begin{array}{lllll} 1 & t_{1} & t_{1}t_{2} & \cdots & \prod_{i=1}^{n-1}t_{i}\\ \frac{1}{t_{1}} & 1 & t_{1} & \cdots & \prod_{i=1}^{n-2}t_{i}\\ \frac{1}{t_{1}t_{2}} & \frac{1}{t_{1}} & 1 & \cdots & \prod_{i=1}^{n-3}t_{i}\\ \vdots & \vdots & \vdots & \ddots & \vdots \\ \frac{1}{\prod_{i=1}^{n-1}t_{i}} & \frac{1}{\prod_{i=1}^{n-2}t_{i}} & \frac{1}{\prod_{i=1}^{n-3}t_{i}} & \cdots & 1 \end{array}\right]$$

The value of the *i*th row and *j*th column in J is $r_{ij}$, meaning that the importance of index $x_{j}$ is $r_{ij}$ times that of index $x_{i}$. The judgment matrix obtained in this way meets the consistency criteria and can be directly used for weight calculation. Finally, the subjective weight of each index can be described as:(10)$$\alpha_{i}={\left(\prod_{j=1}^{n}r_{ij}\right)}^{1/n}/{\sum_{i=1}^{n}\left(\prod_{j=1}^{n}r_{ij}\right)}^{1/n}$$
where $\prod_{j=1}^{n}r_{ij}$ is the product of all elements in line j of J.

#### 2.3.2. The CRITIC Method

Aiming at studying the influence of factor numerical characteristics on weight, the CRITIC method is utilized. The CRITIC weighting method, based on index value, was proposed by Diakoulaki et al. [37]. They discuss the contrast intensity of each index and the conflict between indexes. Through mathematical synthesis, the importance of an index can be objectively measured.

Contrast intensity refers to the difference between different samples within the same index. The contrast intensity is quantified by the standard deviation of the factor as:(11)$$C1_{i}=\sqrt{\sum_{j=1}^{n}{\left(z_{ij}-{\bar{z}}_{j}\right)}^{2}/\left(n-1\right)}$$
where $z_{ij}$ is the *j*th sample value *i*th index, and ${\bar{z}}_{i}$ is the sample mean value of the *i*th index.

Conflict is the degree of difference between indexes. The absolute value of the correlation coefficient is used to calculate the value of conflict as:(12)$$C2_{i}=\sum_{j=1}^{n}\left(1-\left|\rho_{ij}\right|\right)$$
where $\rho_{ij}$ is the correlation coefficient between the *i*th index and the *j*th index.

The greater the contrast intensity and conflict, the more information they contain, meaning the corresponding index has a larger weight value. Thus, the contrast intensity and conflict are used to calculate the point cloud feature index weights determined by the CRITIC method according to the factor numerical characteristics. It can be described as:(13)$$\beta_{i}=C1_{i}*C2_{i}$$

## 3. The Methodology

The process of the MIWSA is shown in Figure 1. It contains four steps: point cloud space division and organization, point cloud feature calculation, feature index weighting, and final simplified point selection. The main computer code is contained in Appendix A.

### 3.1. Point Cloud Space Division and Organization

Although a kd-tree can increase the efficiency, it takes up memory space. The index pointer data, for abundant points, increase the depth of a kd-tree. This leads to high search computation complexity [38]. Thus, the organization that combines a bounding box and a kd-tree is constructed, from which the neighborhood and leaf nodes are determined.

First, a space bounding box is established for the whole point cloud. The pre-set number l should be around the number of neighborhood points in the following operation.

Second, the local kd-tree is constructed. Each kd-tree is constructed in a total of 27 3 × 3 × 3 cubes. The neighborhood points of each point are found in the kd-tree constructed in the cube where the point is and the surrounding 26 cubes. The size of the kd-tree leaf node has an influence on the subsequent simplified points selection.

After the disposition, the point number for each kd-tree is N/F, reducing the depth of the kd-tree effectively. After that, according to Equations (6)–(8), the computation complexity of creating a kd-tree for all points, neighborhood depth-first search, and worst backtracking can be expressed, respectively, as:(14)$$T_{c}=O\left(F\times N+K\times N\times \log N\right)$$
(15)$$T_{d}=O\left(F\times N+27\times N\times \log\left(N/F\right)\right)$$
(16)$$T_{w}=O\left(F\times N+27\times d\times N^{\left(2-1/K\right)}\times F^{\left(1/K-2\right)}\right)$$

Because F tends to be much larger than 27, it is obvious that the complexity is cut down due to the reduction in tree depth, especially for the backtracking computation. In addition, memory space is saved. As for the accuracy, the MIWSA ensures exactness by searching neighborhood points in the surrounding cubes. Former methods tend to contain errors when dealing with points in the marginal region of the first-level division space [39].

### 3.2. Point Cloud Feature Extraction

Five feature indexes are used in the paper, among which the existing calculation formulas are used for the edge and density feature index without change, while the curvature and terrain complexity feature indexes have some modifications in their computation. Moreover, a new feature index is defined to describe the relative point location feature.

#### 3.2.1. Curvature Feature Index

To reduce the calculation process, the MISWA improves the PCA method to estimate the point cloud sharp feature. It was proposed by Hoppe [40] to estimate the normal vector of scattered point clouds using the PCA method.

A covariance matrix is constructed based on local neighborhood data points. Let the coordinate of a point in the neighborhood of point j be $P_{j}$ while its neighborhood gravity center is $O_{j}$, then the covariance matrix can be expressed as:(17)$$Q=\frac{1}{m}\left[\begin{array}{l} P_{j}^{1}-O_{j}\\ \vdots \\ P_{j}^{m-1}-O_{j}\\ P_{j}^{m}-O_{j} \end{array}\right]{\left[\begin{array}{l} P_{j}^{1}-O_{j}\\ \vdots \\ P_{j}^{m-1}-O_{j}\\ P_{j}^{m}-O_{j} \end{array}\right]}^{T}$$

The covariance matrix Q is obviously a symmetric and positive semidefinite matrix, which reflects the geometric information of the local neighborhood surface. The Jacobian method is used to solve the covariance matrix and obtain its eigenvalues $\omega_{1}$, $\omega_{2}$, and $\omega_{3}$, corresponding to eigenvectors $\varphi_{1}$, $\varphi_{2}$, and $\varphi_{3}$, respectively. We can use these values to estimate the curvature of the surface. If one eigenvalue $\omega_{i}$ is smaller than the other two, its corresponding eigenvector can be recognized as the normal vector n. The greater the included angles between the normal vectors of one point and its neighborhood point, the larger the curvature value of this point. Thus, the mean included angles of the eigenvectors are used as the curvature feature index:(18)$$A_{j}=\frac{1}{m}\sum_{i=1}^{m}\left(\frac{n_{j}\cdot n_{i}}{\left|n_{j}\right|\left|n_{i}\right|}+1.1\right)$$
where $n_{j}$ and $n_{i}$ are the normal vectors for points j and i, respectively. The number 1.1 is added to avoid negative values, which may affect subsequent calculations. This formula requires no calculation of included angle, which reduces the computational amount, and can ensure the linear relationship between the calculated result and the included angle of the normal vectors.

#### 3.2.2. Density Feature Index

From the mean distance between point j and its neighborhood points [28], the density feature index is calculated as:(19)$$B_{j}=\frac{1}{m}\sum_{i=1}^{i=m}\left|R_{j}-R_{i}\right|$$

#### 3.2.3. Edge Feature Index

The edge feature index is written as the distance between each point and its neighborhood center of gravity [41] as:(20)$$C_{j}=\left|R_{j}-\frac{1}{m}\sum_{i=1}^{m}R_{i}\right|$$
where m is the number of neighborhood points and $R_{i}$ and $R_{j}$ are the coordinate vectors of points i and j, respectively.

#### 3.2.4. Terrain Feature Index

For terrain point clouds, an index describing the terrain complexity is necessary to evaluate the point importance. This paper uses an index developed from the method of Xu et al. [42]. It is given by:(21)$$D_{j}=\frac{\frac{1}{m}\sum_{i=1}^{m}\left|H_{i}-\bar{H_{j}}\right|}{\frac{1}{N}\sum_{i=1}^{N}\left|H_{i}-H\right|}$$
where $H_{i}$ is the height of point j, $\bar{H_{j}}$ is the average height of neighbor points of point j, and H is the average of all the points. The index reveals the local area undulating characteristics.

#### 3.2.5. 3D Feature Index

The above feature indexes mostly analyze the surface features of point clouds, but the obtained 2.5-dimensional (namely, 2.5D) [43] features are not enough when scanning targets with high vertical complexity such as 3D models, dense buildings, or lush jungles. Hence, a new index is designed for this. As shown in Figure 2, the index calculates the average value of the included angle between vectors from point j to points $i_{1}$, $i_{{}_{2}}$, …, $i_{l}$ in its neighborhood so as to judge the particularity of a point in space. The smaller the average value of the included angles, the more special the point is. Conversely, the larger the angle, the more the point can be represented by the surrounding points.

The index is written by:(22)$$E_{j}=\frac{2}{C_{m}^{2}}\sum_{v_{a},v_{b}\in V,a\neq b}\left(\frac{v_{a}\cdot v_{b}}{\left|v_{a}\right|\left|v_{b}\right|}+1.1\right)$$
where V is the set of all the vectors from point j to its neighborhood points and $v_{a}$ and $v_{b}$ are a pair of different vectors in V. The number 1.1 has the same function as in Equation (22).

### 3.3. Feature Indexes Weighting

Afterwards, the weights of the indexes are calculated by the AHP method and the CRITIC method. Based on Equations (10) and (13), the weights are integrated as:(23)$$\omega_{i}=\frac{\alpha_{i}\beta_{i}}{\sum_{i=1}^{n}\alpha_{i}\beta_{i}}$$
where $\omega_{i}$ is the final weights of indexes A, B, C, D, and E for each point, respectively. Then, the importance quantification result of point j is:(24)$$Z_{j}=\left[A_{j},B_{j},C_{j},D_{j},E_{j}\right]{\left[\omega_{A},\omega_{B},\omega_{C},\omega_{D},\omega_{E}\right]}^{T}$$

### 3.4. Final Simplified Points Selection

The simplified points are selected according to whether they are classified as feature or non-feature points.

First, feature points and non-feature points are determined by sorting the Z values of all points. Points with larger Z values are feature points and are saved as simplified points, and those with smaller values are non-feature ones. Second, according to the kd-tree constructed in the previous process, it is judged whether there are feature points in each leaf node. If not, we select the non-feature point that is closest to the center of gravity of the node to join the simplified points. Finally, a simplified point cloud is obtained.

## 4. Experiment and Discussion

This article focuses on 3D models and field area point clouds. Thus, two experiments were designed to verify the effects of the MIWSA for the two kinds of point clouds. Each experiment utilized two sets of data. We compared the proposed algorithm with the following existing algorithms: the random algorithm [44], the normal algorithm [27], the voxel algorithm [45], the k-means clustering algorithm [23], and the curvature algorithm [30].

In the experiment involving 3D model point clouds, the effects were checked by qualitatively comparing what the results look like after point cloud simplification. Then, we encapsulated the surfaces and quantitatively compared the surface areas and patch numbers of results from different algorithms. Comprehensive evaluations of these algorithms were acquired.

In the field area point cloud experiments, we compared the proposed MIWSA with the five abovementioned algorithms. The simplified results were compared in a preliminary analysis. After that, we created the digital elevation models (DEM) by fitting the original and simplified point clouds. These DEMs were used to accurately evaluate the simplification property of the algorithms.

### 4.1. 3D Model Point Cloud Simplification Test

The dragon dataset (Appendix A) and buddha dataset (Appendix A) released by Stanford University were used for the 3D model point cloud simplification test.

After calculating the feature indexes, the scale values for these indexes in the AHP method are determined. We describe the results as follows. The bending characteristics of the 3D model surface tend to be the most important, so the curvature ranks first. At the same time, we found that the stereoscopic attributes of points are prominent, and the points are distributed on the surface of the model object. Consequently, the angle and edge feature indexes follow. The density feature is not important by comparison. Overall, the descending order of characteristic parameters according to importance is: the density index, the edge index, the angle index, and the curvature index. The scale values between the indexes are shown in Table 2. Each scale value means the corresponding feature index’s multiple over its left one. The farther to the left, the larger the index value.

#### 4.1.1. Point Clouds Simplification Results

The original point cloud and simplified results of the buddha dataset at a simplification rate of 65% are shown in Figure 3. It can be seen from Figure 3a that the buddha has wrinkles on its hemline, which are the sharp feature areas. After simplification, the result of the MIWSA has the best balance between keeping the feature areas prominent and maintaining the integrity of the whole point cloud. In contrast, the normal algorithm and curvature algorithm preserved too many points in feature areas and led to some hollow areas. Meanwhile, the voxel algorithm did not perform well when retaining the feature areas despite avoiding empty areas. For the random algorithm, it also led to some cavities due to its uniform selection in some sparse areas of the point cloud. The k-means algorithm result appears more appropriate, but it still has weaknesses in some convex areas.

Figure 4 shows the original and simplified point clouds at a 65% simplification rate of the dragon dataset. It is clear that there is winding in all parts of the dragon, especially on its back and claws. Compared with other algorithms, the MIWSA can maintain the uniformity of the overall point cloud, and at the same time, it has a better extraction effect on areas with more complex model changes and obvious characteristics. In contrast, the random algorithm was greatly affected by the density of the point cloud, and there are too many reserved points in some areas. The voxel algorithm is insufficient for feature extraction. There are some holes in the results of the curvature and normal algorithms, and their effect is not as good as that of the MIWSA algorithm.

#### 4.1.2. Model Encapsulation Results

To compare the point cloud simplification results more clearly, the point clouds were surface encapsulated, and the results for the buddha dataset and dragon datasets are shown in Figure 5 and Figure 6, respectively. The blue areas in the figures are the surface of the encapsulated model, and the yellow ones are the hollow parts.

It can be seen that for the buddha dataset, the encapsulated surface of the MIWSA has fewer hollows than the other methods, and the representation is more refined at the bending sleeves and skirts. For the dragon dataset, although the surface corresponding to the MIWSA has a certain degree of distortion and deviation in the head and the back of the neck of the dragon, there are basically no cavity areas and few abnormal model surface areas.

Aiming at quantitatively analyzing the accuracy of the model, the surface areas of the model surfaces encapsulated from the simplified point clouds are calculated. The larger the area, the more microscopical the representation of the encapsulated surface. Therefore, the accuracy of the simplified point clouds can be measured.

The comparisons of the surface areas corresponding to the algorithms under different simplification rates are shown in Figure 7a,c. It can be seen that with a reduction in the simplification rate of point clouds, the surface area increases. For the buddha dataset, the surface area corresponding to the result of the MIWSA is 5% to 15% larger than those of the other algorithms. For the dragon dataset, the comparative advantage is not that large, but the value of the MIWSA still occupies first place, which is 0.0002–0.0004 m^2^ larger when compared with the other algorithms.

Similar to the surface area, the patch number of the 3D model can also be used to quantitatively indicate the simplification accuracy of point clouds. The larger the patch numbers, the more precise the simplified results. The patch numbers for the point clouds of the two datasets are shown in Figure 7b,d. In order to clearly express the large values, they are processed by subtracting the minimum value from all the values for each simplification rate. Figure 7b shows the patch numbers of the buddha dataset. It can be seen that the patch number of the MIWSA is always the largest. The lead number above the second largest value ranges from 400 to 1200. As for the dragon dataset, the lead is around 150.

Combined with both the processed point clouds and encapsulated surfaces, it can be concluded that the MIWSA has higher accuracy than the other algorithms for 3D model point clouds. It is able to describe the simplified point clouds in more detail by keeping or abandoning the points using a more appropriate strategy.

#### 4.1.3. Simplification Efficiency

In order to compare the algorithms’ efficiency, we carried out point cloud simplification experiments on the same platform. The type of CPU was Ryzen 4800H (manufactured by Lenovo in China) and the memory was 16 GB. The running time for each algorithm for these point cloud datasets can be seen in Table 3.

It is obvious that the MIWSA has an edge over the normal, k-means, and curvature algorithms in terms of calculating time, which is about 60%, 80%, and 3% of these three algorithms, respectively. Though the random and voxel algorithms can simplify points more quickly, these two can hardly take the points’ geometric and spatial features into account, making their simplification performance terrible.

### 4.2. Field Area Point Cloud Simplification Test

To test the simplification ability of MIWSA for field area point clouds, datasets Samp52 (Appendix A) and Samp53 (Appendix A) released by the International Society for Photogrammetry and Remote Sensing were used.

The order of indexes and their scale values in AHP were determined according to the following analysis. In the field areas in Samp52 and Samp53, the undulating characteristic of the terrain is the key factor for the simplification of point clouds. Thus, the terrain feature index is the most important. For the local parts not covered by vegetation, it is necessary to collect bending information about the object surface, making the angle feature index the second most important. Meanwhile, for wood-covering areas with hierarchical features, it is necessary to extract the feature points inside the cluster point cloud, which means the curvature feature index ranks next. Additionally, the edge and density feature indexes are both not influential in these kinds of point clouds. In general, the terrain complexity index, the angle index, and the curvature index are ranked in the top three. Meanwhile, the density index and the edge index are the last two. The index values and scale values are shown in Table 4.

#### 4.2.1. Point Cloud Simplification Results

Figure 8 illustrates the original and simplified point clouds at an 80% simplification rate. As is shown in Figure 8a, the original point cloud of Samp52 mainly includes gentle slopes, cliffs, rivers, and trees. It can be seen from Figure 8b that the MISWA result reserves more points in the cliffs, trees, and other bump regions, while points in flat areas and slopes are reduced greatly. In contrast, the results of the other algorithms have shortcomings either when extracting features or retaining the overall shape. This reveals that the MISWA has the character of choosing relatively important points and abandoning points that can be replaced by other points without significantly affecting the accuracy.

The original point cloud of Samp53 is shown in Figure 9a, in which cliffs and stepped terrain features can be seen. In addition, there are some wavy parts distributed on the plain. The simplification rate is 80%. The normal algorithm and curvature algorithm choose the points with local special characteristics. However, these characteristics are not suitable for geography point clouds, making these simplified points imprecise. The voxel algorithm and k-means algorithm results retain the overall shape of the point cloud, but they do not emphasize the key features. In comparison, the MIWSA not only extracts the feature points in areas with changing topography but also keeps the dataset integrated.

#### 4.2.2. DEM Construction Results

In order to evaluate the accuracy of the simplification results for field area datasets more intuitively, a method based on geographical analysis is proposed in this subsection.

One of the uses of terrain point clouds is to construct digital elevation models (DEMs) with which this paper evaluates the algorithm accuracy. Grid DEMs are generated by fitting the original and the simplified point clouds. Then, the elevation values of the corresponding position grids in the two models are made different to indicate the simplification error reflected in each grid. By analyzing the overall spatial distribution and statistical characteristics of the errors, the point cloud simplification results can be evaluated.

Figure 10 shows the elevation difference distributions of Samp52 at a simplification rate of 80%. The errors are more remarkable along the river for all results, among which the MIWSA keeps both the extreme values and overall values to the minimum.

As for Samp53, Figure 11 shows the elevation difference distributions. It can be seen that although the MIWSA leads to more errors in the plain to some extent, there are significantly fewer errors in the steep terrain areas than there are when using the other algorithms. This reflects that the MIWSA properly adjusts the strategy of retention and abandonment for points to raise the point cloud simplification accuracy in general.

In order to assess the precision of the simplified results quantitatively, we calculated two indicators, namely the mean value $Y_{mean}$ and root mean square value $Y_{rms}$ of the different values in the grids. $Y_{mean}$ and $Y_{rms}$ are given by:(25)$$Y_{mean}=\frac{1}{s}\sum_{i=1}^{s}Z_{i}$$
(26)$$Y_{rms}=\sqrt{\frac{1}{s}\sum_{i=1}^{s}{\left(Z_{i}-Y_{ave}\right)}^{2}}$$
where s is the number of grids and Z is the elevation difference in each grid.

Figure 12 illustrates the results at simplification rates from 80% to 50% at 5% intervals. Figure 12a,b corresponds to Samp52, and Figure 12c,d corresponds to Samp53.

For Samp52, the MIWSA has the smallest indicator values other than at a rate of 80%. With a reduction in the simplification rate, the advantage of the MIWSA becomes greater. Overall, the indicator values of the MIWSA are 0–80% smaller than those of other algorithms. Among the other algorithms, the k-means algorithm is the best, while the random algorithm performs the worst.

For Samp53, the MIWSA has the best effects at all the simplification rates. Its smallest $Y_{mean}$ value difference with other algorithms is around 0.01 m, while its greatest difference is 0.1 m. In addition, the $Y_{rms}$ value of the MIWSA is 2–15% smaller. As the simplification rate declines, the advantage of the proposed algorithm also becomes prominent.

Generally, the MIWSA has good adaptation for field area point clouds. Compared with the existing algorithms, it is able to better maintain the features and the overall shape. By quantitative assessment, the results of the MIWSA algorithm are less different from the original point cloud, and the error distribution is more uniform. The MIWSA is able to maintain a small numerical error value and a narrow distribution range.

#### 4.2.3. Simplification Efficiency

As in the 3D model point cloud simplification experiment, the algorithms’ efficiency is compared. The running time can be seen in Table 5.

As shown in Table 5, the MIWSA running times are about 30% and 50% shorter than the normal algorithm. The k-means algorithm is much slower. The curvature algorithm’s performance is close to that of the MIWSA. The random and voxel algorithms are comparatively quicker due to their simple principle.

## 5. Conclusions

To solve the issues of computation cost and storage space occupation caused by large point clouds collected during 3D scanning, this paper proposes a novel point cloud simplification algorithm, MIWSA, based on point features and spatial relationships. First, the point cloud is organized via a bounding box and a kd-tree for subsequent neighborhood searching and point division. Second, five feature indexes are calculated to reveal the characteristics from different perspectives: point cloud shape, density, object edge information, terrain information contained, and inner relationship. Third, the AHP and CRITIC methods are utilized to give the weights of the feature indexes, and a comprehensive index is thus attained. Fourth, each point is classified as a feature point or non-feature point according to the comprehensive index value. The former are saved into final simplified points, and the latter are determined to be saved or abandoned according to their location relationship with the feature points. Overall, the MIWSA is able to extract multifaceted points by considering both the geometric attributes and the spatial relative position.

In order to verify the effectiveness of the MIWSA, experiments using 3D model point clouds and field area point clouds are carried out. Compared with five existing algorithms, the MIWSA has superior simplification effects in both quantitative and qualitative terms. For 3D model point clouds, the MIWSA is able to maintain the details and contour of the object well and retains the largest surface area and the greatest patch number. For field area point clouds, the MIWSA algorithm can reduce the number of errors and keep the error distribution more uniform. By adjusting the scale order and values according to the point cloud scenes, the proposed algorithm can flexibly simplify different point clouds. In terms of efficiency, the MIWSA operates more quickly than the normal, curvature, and k-means algorithms and more slowly than the random and voxel algorithms. When comprehensively considering the effect, adaptation, and efficiency, the MIWSA is better than the other algorithms.

The MIWSA should be tested on more point cloud datasets. Meanwhile, the AHP weighting method relies on subjective analysis. Thus, it requires the acquisition of a detailed summary of weight distributions for different types of point clouds.

## Figures and Tables

**Figure 1 sensors-22-07491-f001:**
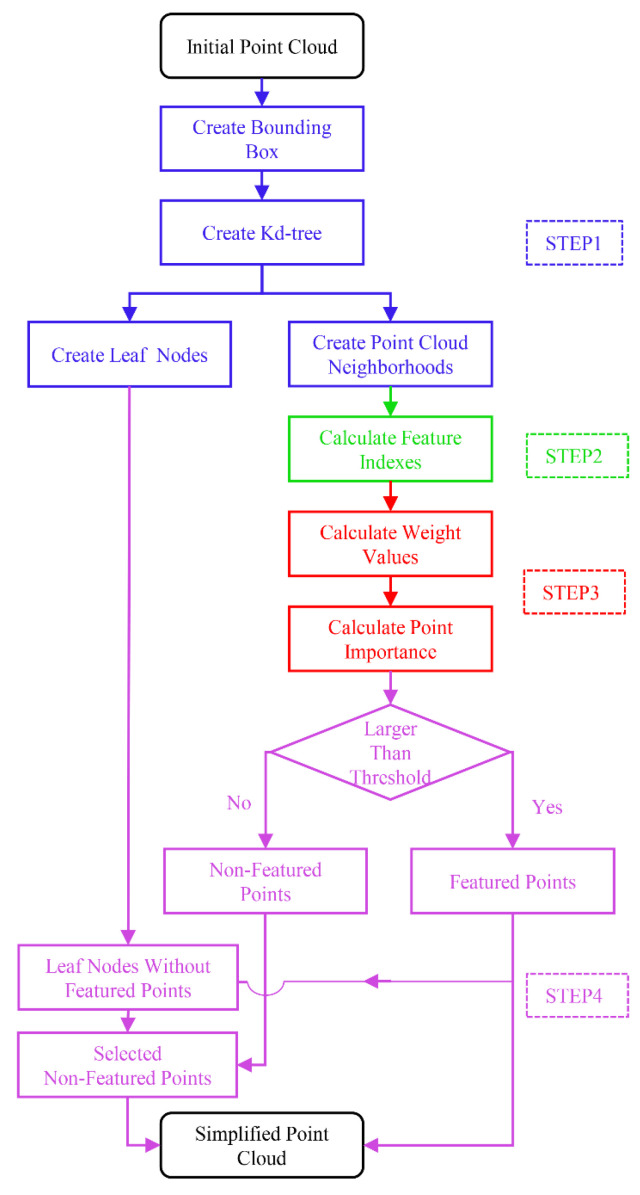
Flow chart of the MIWSA.

**Figure 2 sensors-22-07491-f002:**
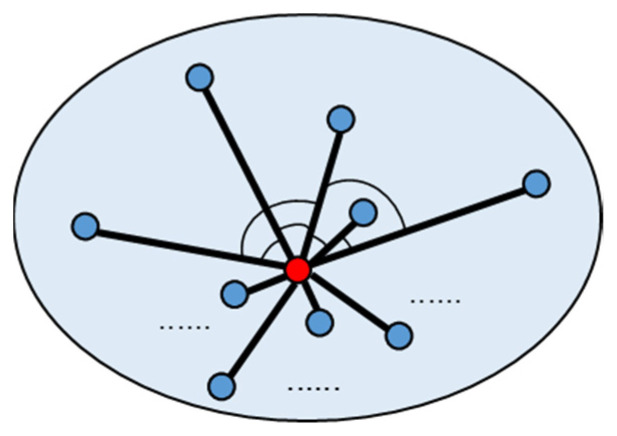
Schematic diagram for 3D Feature Index.

**Figure 3 sensors-22-07491-f003:**
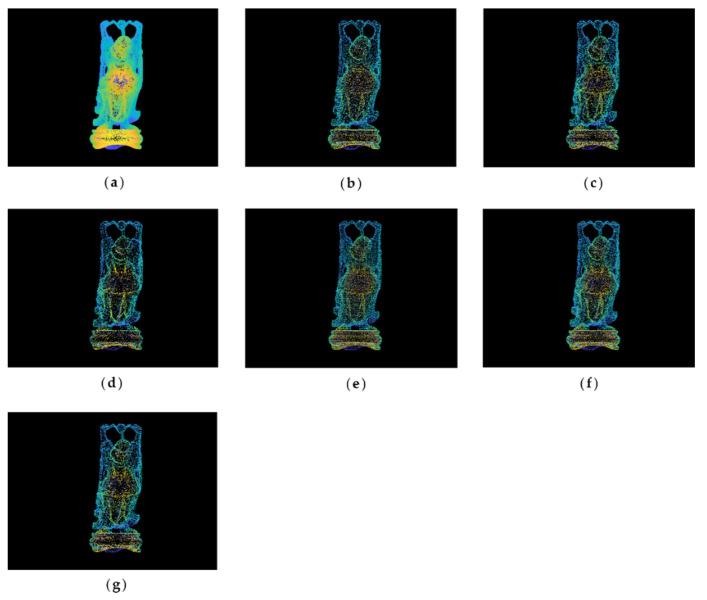
Original and simplified point clouds of the buddha dataset: (**a**) original point cloud; (**b**) MIWSA; (**c**) random algorithm; (**d**) normal algorithm; (**e**) voxel algorithm; (**f**) k-means algorithm; (**g**) curvature algorithm.

**Figure 4 sensors-22-07491-f004:**
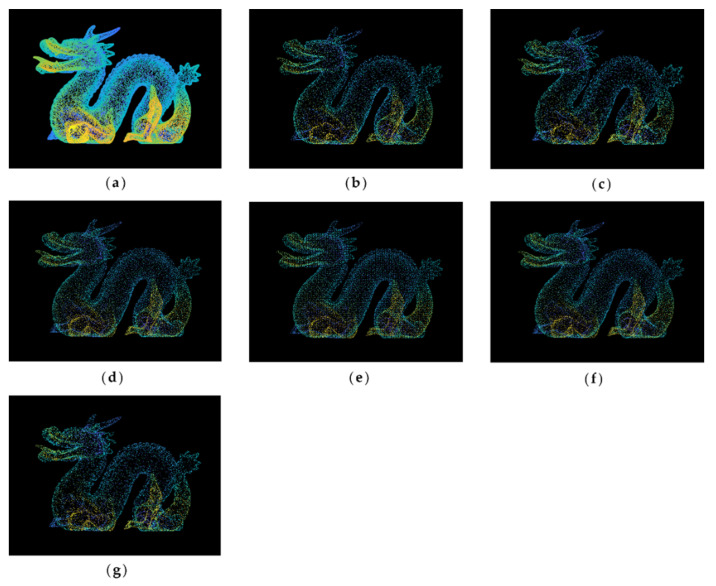
Original and simplified point clouds of the dragon dataset: (**a**) original point cloud; (**b**) MIWSA; (**c**) random algorithm; (**d**) normal algorithm; (**e**) voxel algorithm; (**f**) k-means algorithm; (**g**) curvature algorithm.

**Figure 5 sensors-22-07491-f005:**
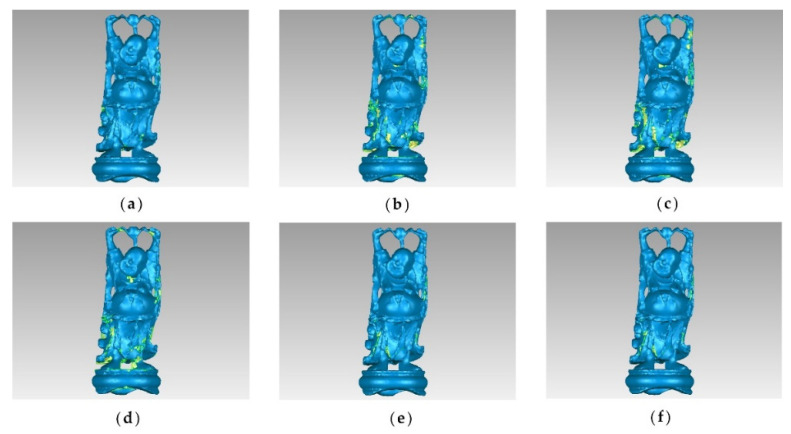
Encapsulated surfaces of point clouds corresponding to the buddha dataset: (**a**) MIWSA; (**b**) random algorithm; (**c**) normal algorithm; (**d**) voxel algorithm; (**e**) k-means algorithm; (**f**) curvature algorithm.

**Figure 6 sensors-22-07491-f006:**
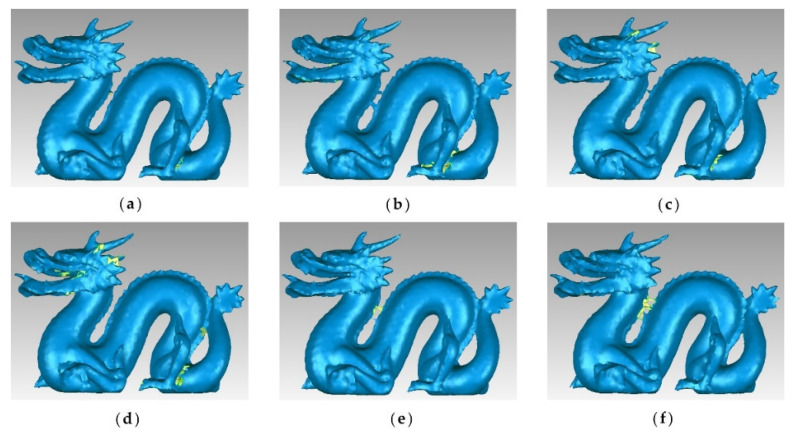
Encapsulated surfaces of point clouds corresponding to the dragon dataset: (**a**) MIWSA; (**b**) random algorithm; (**c**) normal algorithm; (**d**) voxel algorithm; (**e**) k-means algorithm; (**f**) curvature algorithm.

**Figure 7 sensors-22-07491-f007:**
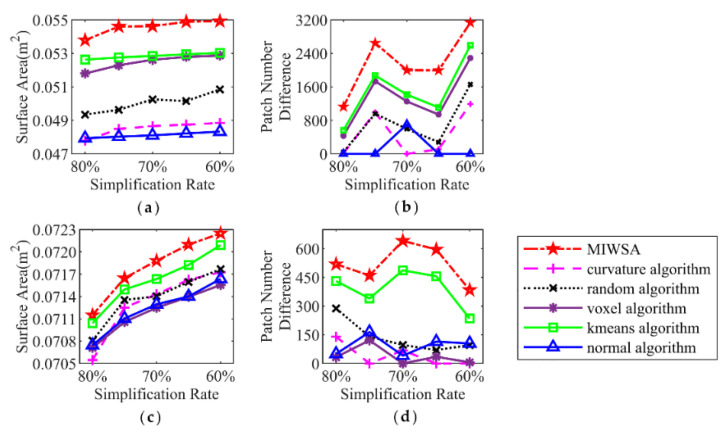
Surface area and patch numbers of the encapsulated surfaces: (**a**) encapsulated surface areas of Samp52; (**b**) encapsulated surface patch numbers of Samp52; (**c**) encapsulated surface areas of Samp52; (**d**) encapsulated surface patch numbers of Samp52.

**Figure 8 sensors-22-07491-f008:**
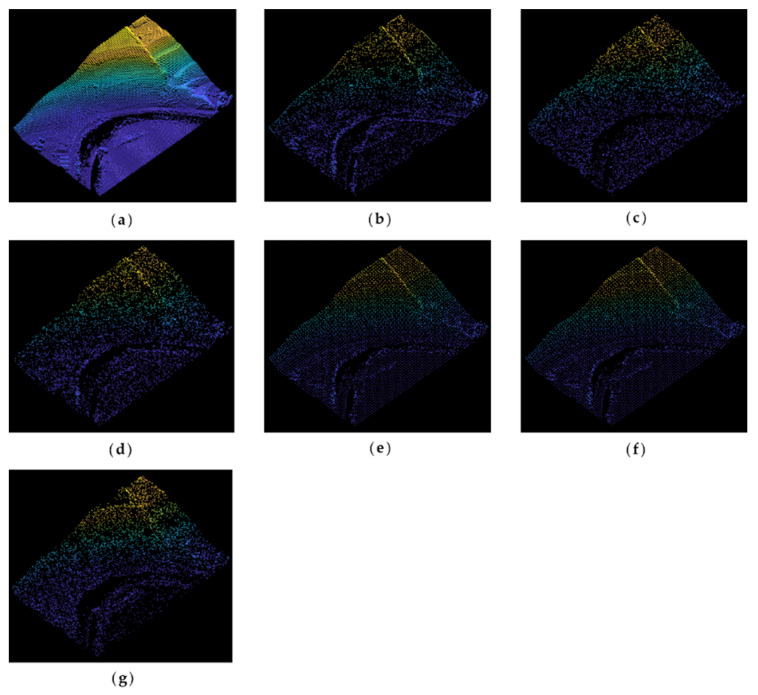
Original and simplified point clouds of Samp52: (**a**) original point cloud; (**b**) MIWSA; (**c**) random algorithm; (**d**) normal algorithm; (**e**) voxel algorithm; (**f**) k-means algorithm; (**g**) curvature algorithm.

**Figure 9 sensors-22-07491-f009:**
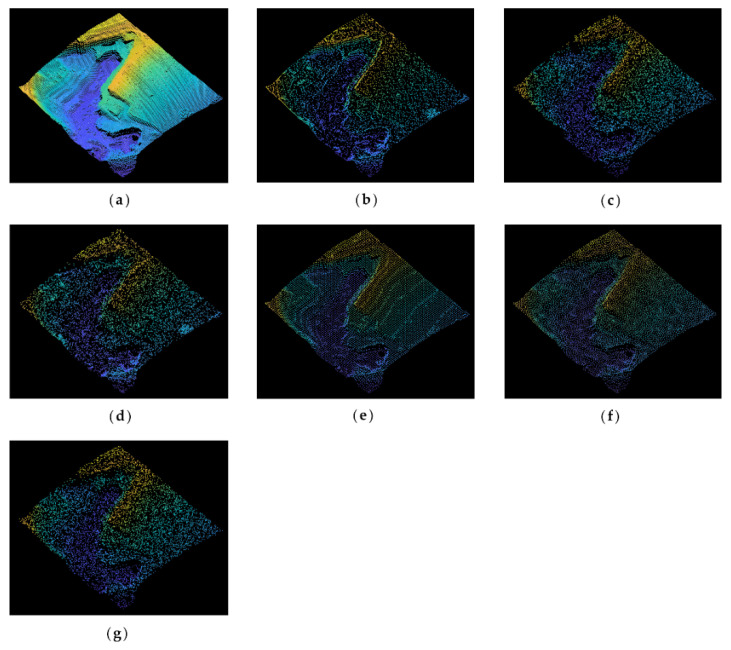
Original and simplified point clouds of Samp53: (**a**) original point cloud; (**b**) MIWSA; (**c**) random algorithm; (**d**) normal algorithm; (**e**) voxel algorithm; (**f**) k-means algorithm; (**g**) curvature algorithm.

**Figure 10 sensors-22-07491-f010:**
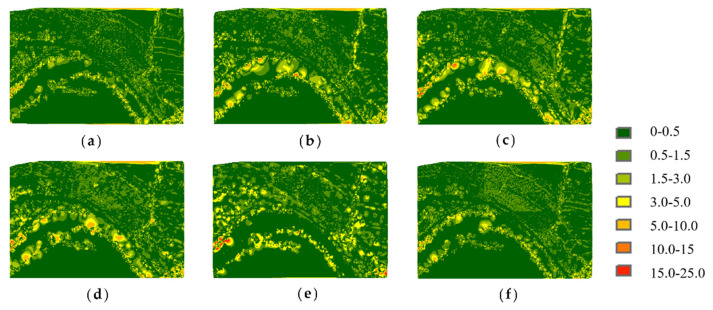
Elevation difference distributions of Samp52: (**a**) MIWSA; (**b**) random algorithm; (**c**) normal algorithm; (**d**) voxel algorithm; (**e**) k-means algorithm; (**f**) curvature algorithm.

**Figure 11 sensors-22-07491-f011:**
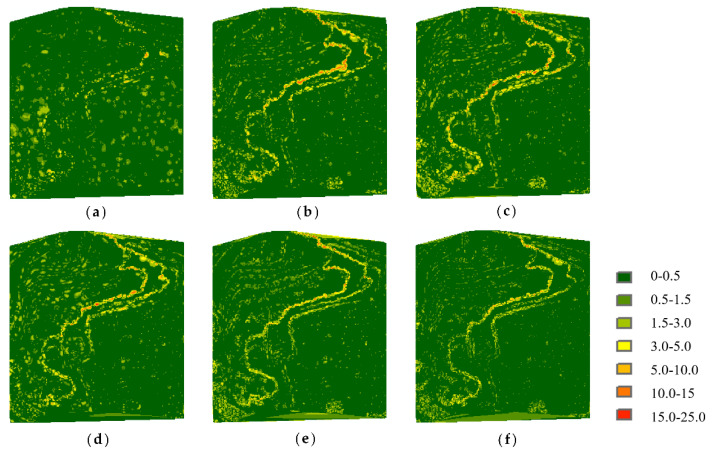
Elevation difference distributions of Samp53: (**a**) MIWSA; (**b**) random algorithm; (**c**) normal algorithm; (**d**) voxel algorithm; (**e**) k-means algorithm; (**f**) curvature algorithm.

**Figure 12 sensors-22-07491-f012:**
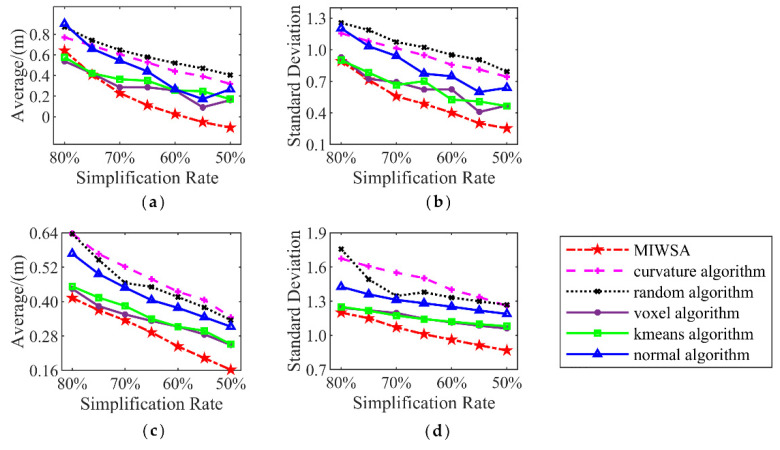
Mean and RMS of elevation difference: (**a**) Mean of elevation difference of Samp52; (**b**) RMS of elevation difference of Samp52; (**c**) Mean of elevation difference of Samp53; (**d**) RMS of elevation difference of Samp53.

**Table 1 sensors-22-07491-t001:** Scale values between adjacent indexes.

Scale Value	Meaning	Scale Value	Meaning
1.0	Equally Important	1.6	Obviously Important
1.2	Slightly Important	1.8	Absolutely Important
1.4	Strongly Important		

**Table 2 sensors-22-07491-t002:** Scale values of indexes for 3D model experiments.

Feature Indexes	Density	Edge	Angle	Curvature
Scale Values		1.2	1.4	1.2

**Table 3 sensors-22-07491-t003:** Point numbers and running time for datasets buddha and dragon (unit: s).

**Dataset**	**Original Point Number**	**Simplified Point Number**	**Normal**	**Random**
buddha	108,731	38,056	23.351	2.379
dragon	87,529	30,635	14.342	2.061
**Dataset**	**Voxel**	**K-means**	**Curvature**	**MIWSA**
buddha	0.093	508.438	15.763	15.312
dragon	0.081	325.871	8.298	6.801

The table forms a new line.

**Table 4 sensors-22-07491-t004:** Scale values of indexes for field area experiments.

Feature Indexes	Density	Edge	Curvature	Angle	Terrain
Scale Values		1.0	1.4	1.2	1.2

**Table 5 sensors-22-07491-t005:** Point numbers and running time for datasets Samp52 and Samp53 (unit: s).

**Dataset**	**Original Point Number**	**Simplified Point Number**	**Normal**	**Random**
buddha	34,378	6875	3.701	0.571
dragon	22,474	4495	2.651	0.485
**Dataset**	**Voxel**	**K-means**	**Curvature**	**MIWSA**
buddha	0.029	167.516	2.491	2.484
dragon	0.024	124.109	1.171	1.388

The table forms a new line.

## Data Availability

Publicly available datasets were analyzed in this study. The dragon and buddha datasets released by Stanford University can be found here: http://graphics.stanford.edu/data/3Dscanrep/, accessed on 27 August 2022. The datasets released by the International Society for Photogrammetry and Remote Sensing can be found here: https://www.itc.nl/isprs/wgIII-3/filtertest/downloadsites/, accessed on 27 August 2022.

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
