# Peer review of "A Point Cloud Simplification Algorithm Based on Weighted Feature Indexes for 3D Scanning Sensors"

_sensors, 2022, doi:10.3390/s22197491_

Round 1

Reviewer 1 Report

Dear Editor / Authors,

In the manuscript the authors studied on a new  algorithm which can be used for reducing the point cloud data.The authors made a good background research and clearly revealed the deficiency of existing algorithms. I think the paper is well researched and analyses were done systemically.  Results are consistent and the paper is well written.

This manuscript potentially has a positive contribution for other researchers.

As the result of the review, the issue given below should be considered;

The number of points for original data and after simplification process for all algorithms should be given in a table.

Author Response

Dear Reviewer: 

The revised manuscript is in the attachment, and here is the response to your comment.

Comments: The number of points for original data and after simplification process for all algorithms should be given in a table.

Response: The point numbers for original data and after simplification process are given in Table 3 and Table 5. Because the simplification rates are same for all algorithms, the point numbers for simplified point clouds are not repeated for each algorithm.

I am eagerly looking forward to hearing from you, because the paper is important for me.

Best Regards

Zhiyuan Shi

Reviewer 2 Report

The manuscript is devoted to the point cloud simplification algorithm obtained using 3D scanning. An improved multi-index weighting algorithm (MIWSA) is proposed. The MIWSA algorithm uses a bounding box and kd-trees to organize points, as well as additional feature indexes to extract features. The experiments have shown that higher accuracy and lower computational complexity are achieved compared to existing algorithms.

These research results are original and have scientific value. Nevertheless, there are certain moments demanding explanations.

1. It is indicated, that the accuracy of the new algorithm is 5-15% higher than the existing algorithms. More attention should be paid to justifying the criteria for assessing accuracy for 2D and 3D simplified point clouds.

2. The new algorithm, according to the authors, has a lower computational complexity. The proposed improvement of the algorithm in terms of accuracy should lead to its complication and, as a result, to an increase in computational complexity. It is desirable to reflect in more detail in the manuscript, due to which an increase in performance is achieved.

3. It is desirable to explain the description of the new D Feature Index with Figure.

4. The analysis of inter-criteria correlation was performed using weight coefficients obtained with the help of expert assessments. The examples, shown in tables 2 and 4, includes 3 and 4 values from 5 indices. Requires additional.

Author Response

Dear Reviewer:

The revised manuscript is in the attachment, and here are the responses to your comments.

Comment 1: It is indicated, that the accuracy of the new algorithm is 5-15% higher than the existing algorithms. More attention should be paid to justifying the criteria for assessing accuracy for 2D and 3D simplified point clouds.

Response: The justification for the criteria is added in the abstract.

Comment 2: The new algorithm, according to the authors, has a lower computational complexity. The proposed improvement of the algorithm in terms of accuracy should lead to its complication and, as a result, to an increase in computational complexity. It is desirable to reflect in more detail in the manuscript, due to which an increase in performance is achieved.

Response: The analysis for the increase in computation performance has been added below Table 3 and Table 5.

Comment 3: It is desirable to explain the description of the new D Feature Index with Figure.

Response: Figure 2 is added to explain the new feature index. And I am sorry for mistakenly write ‘3.D Feature Index’ as ‘3. D Feature Index’. It has also been revised.

Comment 4: The analysis of inter-criteria correlation was performed using weight coefficients obtained with the help of expert assessments. The examples, shown in tables 2 and 4, includes 3 and 4 values from 5 indices. Requires additional.

Response: The two examples respectively selected 4 and 5 indices. And the paragraph above table 2 interpreted that in Table 2 and Table 4, ‘Each scale value means the corresponding feature index’s multiple over its left one. The farther to the left, the larger the index value.’. Thus, there is no require for additional contents.

I am eagerly looking forward to hearing from you, because the paper is important for me.

Best Regards

Zhiyuan Shi
